# Study and Analysis of the Effect of Polyamide Seamless Knitted Fabric with Different Graphene Content on the Blood-Flow Velocity of Human-Skin Microcirculation

**DOI:** 10.3390/ma15196853

**Published:** 2022-10-02

**Authors:** Yixing Zhang, Zimin Jin, Jiaxue Chen, Mingtao Zhao, Yuqiang Sun, Yijing Song

**Affiliations:** 1College of Textile Science and Engineering, Zhejiang Sci-Tech University, Hangzhou 310018, China; 2Zhejiang Bangjie Holding Group Co., Ltd., Yiwu 322009, China; 3College of Life Sciences and Medicine, Zhejiang Sci-Tech University, Hangzhou 310018, China; 4Guangzhou Jinrui Technology Co., Ltd., Guangzhou 510220, China

**Keywords:** graphene polyamide, seamless knitting, promotion multiple of blood flow on human-surface-skin microcirculation

## Abstract

In this paper, four kinds of polyamide yarns with different graphene contents and three kinds of seamless knitting structures were used. The scheme of samples was established according to the comprehensive experimental design method, and 12 pieces of knitted fabric samples were woven on the seamless knitting machine. Through testing and analyzing the influence of each sample on the blood-flow rate of human-surface-skin microcirculation, the research shows that the higher the content of graphene in the veil, the better the promotion effect of the fabric prepared under this process condition on the blood-flow rate of human-surface-skin microcirculation. Sample 11# with the veil type of GP-0.8% and fabric weave of 1+1 simulated ribbed stitch has the strongest effect in this experiment, with a promotion multiple of 1.2189, and the influence of tissue structure is not obvious. The relevant performance test data and experimental research results in this paper provide empirical data support for developing medical or health textiles related to promoting the blood-flow velocity of skin microcirculation.

## 1. Introduction

Periarthritis of the shoulder (frozen shoulder for short) is a chronic, specific inflammation in which shoulder pain is progressively aggravated and accompanied by limited mobility of the shoulder joint [1]. Regarding the treatment of frozen shoulder, at present, most treatments are administered internally, but their efficacy is not good. The use of external treatment can be directly reached in a hospital, and the effect is good. At present, there are many controlled experimental studies and clinical applications of traditional infrared therapy and comprehensive infrared therapy as external treatment methods. Zhao used LD-AF digital low intermediate frequency far-infrared warmer with traditional Chinese medicine to treat 96 cases of scapulohumeral periarthritis and achieved satisfactory results [2]. Wang et al. treated 67 cases of scapulohumeral periarthritis with acupuncture combined with multi-functional far-infrared local irradiation and functional exercise and achieved satisfactory results [3]. Fatma et al. concluded that technological progress provides new technologies for transmitting far-infrared radiation to the human body, and professional lamps and saunas that provide pure far-infrared radiation have become safe, effective, and widely used light sources to produce therapeutic effects [4]. Graphene fiber is an excellent textile material that can emit far-infrared rays [5]. The far-infrared biological effect of fiber materials, when it acts on human skin, can improve the local blood-flow rate of superficial skin, and at the same time, it can also act on the biological body to improve metabolism [6]. Therefore, the application of this kind of graphene fiber material in clothing fabrics can improve the far-infrared performance of the fabric to a certain extent and improve the promotion of the blood-flow rate of human-skin microcirculation by the contact coverage of the fabric [7]. To a certain extent, the blood-flow velocity of human-surface skin can evaluate microcirculation and characterize the effects of far-infrared rays on the human body [8,9]. 

In this paper, graphene polyamide yarns were combined with seamless knitting technology. We explored the influence of seamless knitting samples prepared from polyamide, with different graphene contents and different tissue structures, on the blood-flow velocity of human-surface-skin microcirculation. In addition, we provided a certain reference value for the application of graphene far-infrared medical care textiles in the medical auxiliary health care of chronic pain diseases such as scapulohumeral periarthritis. If the test and research prove that this material has an exact effect on promoting the blood-flow rate of skin microcirculation, patients with periarthritis of the shoulder will be treated by wearing this product directly, eliminating the trouble of going to the hospital regularly for infrared light irradiation treatment, and it will be worth promoting for use in clinical practice.

## 2. Materials and Methods

### 2.1. Materials

To explore the effects of graphene content and tissue structure on the properties of seamless knitted fabrics, four kinds of polyamide yarns with different graphene contents were selected, and three different tissue structures were designed. The sample weaving scheme was established according to the comprehensive experimental design method, and a total of 12 samples were prepared by SM8-TOP2 electronic seamless forming knitting circular machine (Santoni (Shanghai) Knitting Machine Co., Ltd. (LONATI Group in Italy), Shanghai, China).

#### 2.1.1. Selection of Yarn Scheme

In this paper, 22.2dtex(20D) polyamide/22.2dtex(20D) polyurethane wrapped yarn provided by Yiwu Huading Polyamide Co., Ltd. was used as inner yarn (Yiwu, China). For veils, 77.8dtex(70D) graphene polyamide yarns with 0.0%, 0.1%, 0.2%, and 0.8% graphene contents were selected. (For the convenience of narration, in this paper, graphene polyamide yarns with graphene contents of 0.0%, 0.1%, 0.2%, and 0.8% were defined as GP-0.0%, GP-0.1%, GP-0.2% and GP-0.8%, respectively.) The specific yarn specifications are shown in Table 1.

#### 2.1.2. Fabric Structure Design

The structure affects the surface-texture morphology and physical properties of the fabric. The commonly used structures in the seamless knitting industry are as follows:Weft Plain Stitch.

The weft plain stitch is one of the basic structures of single-sided weft knitted fabrics [10]. The fabric woven by this stitch is very tight and has a smooth front and good extensibility. It is often used to prepare underwear, socks, and other products.

2.Rib Stitch.

The rib stitch is also one of the basic structures of weft knitted fabrics [11]. It is composed of front and back coils arranged in a certain longitudinal proportion. The fabric woven from this stitch has a medium tightness. This kind of fabric has great extensibility when applied with transverse tension and can be used to prepare clothes with good elasticity, such as yoga clothes. The common rib fabrics in the market include 1+1 simulated rib and 2+2 rib.

3.Simulated Rib Stitch.

The fabric woven by the simulated ribbed stitch is looser and has a longer floating thread on the back of the fabric, while the front of the fabric presents appearance characteristics similar to ribbed fabric. Compared with plain knitted fabric and ribbed fabric, it has a more thermal effect and is often used in the production of socks.

Considering the product and how its performance is affected by the closeness of the structure, we selected three levels of the stitch of weft knitted fabrics that are commonly used in the seamless knitting industry and are in the tight, medium, and loose levels, as shown in Figure 1.

#### 2.1.3. Establishment of Sample Scheme

In this paper, polyamide yarns with different graphene contents were used as the veil type, and we studied the influence of fabric technology on the blood-flow rate of human-surface-skin microcirculation according to four veil types and three tissue structures.

To comprehensively analyze the fabric properties of different veils and different tissue structures as experimental factors, according to the multi-factor and multi-level tests of this test, the four levels of different veil types were denoted as factor A (GP-0.0%, GP-0.1%, GP-0.2%, and GP-0.8%), and the three levels of tissue structure were factor B (weft plain stitch, 1+1 simulated rib, and 1+3 simulated rib). Twelve samples were woven according to the comprehensive experimental design method. The sampling scheme is shown in Table 2.

### 2.2. Methods

#### 2.2.1. Experimental Equipment and Test Principle

Laser Speckle Contrast Imaging (LSCI), a non-contact testing technology for testing the blood flow of human-surface-skin microcirculation, has been widely used in clinical research in recent years because of its high sensitivity and accuracy [12,13]. LSCI is a non-contact imaging technology based on the principle of speckle contrast analysis. Speckle occurs because large amounts of light interfere with each other due to scattering and diffuse reflection when a laser beam irradiates an object with a rough surface. At this time, if the rays are superimposed with each other, bright spots will appear. Such images are called laser speckle patterns.

LSCI has excellent temporal and spatial resolution and has the advantage of full-field measurement. At present, this technology has been applied in many aspects, such as cerebral-blood-flow monitoring and skin-microcirculation monitoring in medicine. Therefore, the use of LSCI is of great significance in measuring medical-related parameters such as blood-flow velocity and microcirculation blood perfusion, studying the pathogenic mechanism in the pathology related to microcirculation perfusion disorder, diagnosing various diseases, and preventing the need for health care and alleviating pain.

The instrument used to test the promotion effect of fabrics on the blood flow of human-skin microcirculation involved in this paper was the BVI microcirculation-blood-flow imager provided by Shenzhen Shengqiang Technology Co., Ltd. (Shenzhen, China). The experimental apparatus, test schematic diagram, and curves of dynamic blood-flow rate were obtained by covering 1+3 simulated rib samples of GP-0.0% and GP-0.2% at two test points of the arm, as shown in Figure 2. Table 3 shows the average values of dynamic blood-flow rate at three periods of two test points, which were intercepted from the curves obtained by covering 1+3 simulated rib samples of GP-0.0% and GP-0.8% at two test points of the arm.

Working Principle of the Instrument.

The instrument is divided into three parts: host, clamp, and support. The host includes a power supply, optical system, electronic processing, and control system. According to the principle of LSCI imaging, when human living tissue is irradiated by infrared light, the hemoglobin in blood vessels contained in the tissue has a more obvious absorption effect on light than surrounding tissues. After conversion by the instrument image processing system, a visible image of veins can be formed on the display screen. The principle advantage of the instrument is that it can capture the blood-flow-velocity imaging of living tissues’ microcirculation within a large area. The largest test area of the instrument is 18 cm × 12 cm, and the laser is launched by the host machine. When the surface of the measured object is uneven, the incident light will backscatter, and the optical path of the light reaching the imaging surface of the camera is different, so random interference will occur on the imaging surface. Thus, speckle patterns with different light and dark effects are produced.

#### 2.2.2. Experimental Materials and Test Methods

Sample Specifications of Wrapping Arm.

According to the arm circumference of the 10 subjects and considering that no pressure should be applied when wrapping the arm of the sample, the size of the sample was set as follows: the width was 10 cm, the length of the test point, C1, near the wrist was 20–23 cm, and the length of the test point, C2, near the elbow was 23–27 cm. To facilitate the test, a 2 cm × 2 cm square hole was set in the middle of each sample. 

2.Pre-Test Preparation Requirements.

Because the test results of blood-flow rate on the human-body-skin microcirculation are easily affected by the physiological factors of the tested person and the external environment, the impact of these factors were avoided as much as possible during the test. Before the experimental test, the samples were humidified for 24 h according to the standard GB/T 6529-2008 [14]. The ambient temperature of the laboratory was controlled at 25 °C and the relative humidity was 65%. The room was kept in a non-ventilated state, and all light aside from the indoor light was shielded. The inside of the non-dominant arm of the tested person was observed in advance and marked with a 2 cm × 2 cm square test point. To ensure the validity of the test data, the tested person’s test position was be contaminated with water within 30 min of the test and they did eat food that may have caused human excitement, including functional drinks such as tea, coffee, etc., with 24 h of the test. The tested person also needed to sit still for 30 min in the test environment to adapt to the environmental conditions. After the tested person was in a calm state, the arm was placed on the test platform for 20 min. The whole process needed to be kept in a static state.

3.Conditions of the Subjects and Test Sites.

To control the influencing factors of the experiment, we increased the number of samples and reduced the impact of individual differences on the experimental results. A total of 10 healthy adult males aged 22–26 years were selected for this study, with a height of 160–170 cm and a weight of 55–70 kg.

Human skin is widely distributed with free nerve endings overlapping each other in a certain range, which makes it highly sensitive. According to relevant studies, the distribution of nerve endings is more extensive in human limbs, which indicates that the skin on the limbs is more susceptible to external stimuli and generates responses [15]. Therefore, the visible superficial veins on the skin surface needed to be avoided as much as possible when testing the blood velocity of microcirculation on the human-skin surface. The test position was selected at least 5 cm away from the wrist and 2.5 cm away from the cubical fossa, and the size of the test position was 2 cm × 2 cm. The marking diagram of the position of the test point on the arm is shown in Figure 3 [16].

4.Test Steps for Blood-Flow Promotion Multiples of Microcirculation on Human-Surface Skin.

Under the above pre-test preparation requirements, we tried to keep the subject in a relatively stable state after they adapted to the laboratory environment.

First, the visible distribution of superficial veins in the arm of the subject was observed, and the test positions of 2 cm × 2 cm were marked with a black marker.

After the arm of the tested person was placed on the test bench for 20 min and reached a stable state, the GP-0.0% sample was covered at test point C1, and the C2 was kept blank. After 20 min, the dynamic blood-flow velocity values of 5 min at C1 and C2 were framed and tested by the computer software(software version number: Laser Microcirculation Winform), and then the original ratio of blood-flow changes on the skin microcirculation was calculated according to the average value of blood-flow velocity in the two areas automatically obtained by the test software. After this test, the subject had to sit still for 10 min for the next test.

Next, we covered the GP-0.0% sample and the GP-0.1% sample at test points C1 and C2, respectively. After 20 min, we repeated the steps of testing the original ratio of blood-flow-rate change to obtain the average value of blood-flow rate after covering the sample at test points C1 and C2 and calculated the ratio of blood-flow change in human-skin microcirculation. After the test, the tested person needed to remain calm for 10 min.

The sample at test point C1 remained unchanged. We replaced the sample at test point C2 with GP-0.2% and GP-0.8%, respectively, and repeated the above test steps.

5.Fabrics Test Scheme.

The test point C1 was covered with GP-0.0% samples, and test point C2 was covered with GP-0.1%, GP-0.2%, and GP-0.8% samples, respectively. The two test points C1 and C2 had to correspond to the samples with the same structure during each test.

We selected two test points on the inner side of the arm of the subject. The state of the uncovered sample and microcirculation-blood-flow imaging are shown in Figure 4(a1,a2). After covering the GP-0.0% sample at C1 for 20 min, we tested the original ratio of blood-flow change in human-skin microcirculation at C1 and C2. The test state and blood-flow imaging are shown in Figure 4(b1,b2). Then, after covering GP-0.1%, GP-0.2%, and GP-0.8% samples in sequence at the C2 test point for 20 min, we tested the change ratio of blood flow of human-skin microcirculation at C1 and C2. The coverage state and blood-flow imaging of the GP-0.1% sample are shown in Figure 4(c1,c2); The coverage state and blood-flow imaging of the GP-0.2% sample are shown in Figure 4(d1,d2); The coverage state and blood-flow imaging of GP-0.8% sample are shown in Figure 4(e1,e2).

6.Calculation of Blood-Flow Promotion Multiple of Human-Surface-Skin Microcirculation.

The effect of fabric on promoting blood-flow velocity of microcirculation on human skin can be shown by the change in blood-flow velocity and the ratio of blood-flow velocity. Refer to the LSCI detection method to test the original ratio of blood-flow change and the ratio of blood-flow change, and take the blood-flow promotion multiple (FhFy) obtained after covering the sample as the evaluation index of the effect of the sample on promoting blood-flow velocity of microcirculation on human skin. The calculation formulas are shown in (1) and (2):(1)$$F=\frac{f_{C1}}{f_{C2}}$$
(2)$$M=\frac{F_{h}}{F_{y}}$$
where $f_{C1}$ is the blood-flow value of microcirculation of human skin at test point *C*1; $f_{C2}$ is the blood-flow value of microcirculation of human skin at test point *C*2; F  is the ratio of blood-flow rate; $F_{y}$ is the original ratio of microcirculatory blood-flow changes; $F_{h}$ is the change ratio of microcirculatory blood flow; and M is the blood-flow promotion multiple after covering the sample.

## 3. Results

According to the above steps, we tested and calculated the blood-flow promotion multiples of microcirculation on the skin of each subject after covering the sample and took the arithmetic average value of the blood-flow promotion multiples of each sample. The results are shown in Table 4. The blood-flow promotion effect of each sample was ranked as follows: 11# > 10# > 12# > 8# > 9# > 7# > 5# > 6# > 4# > 1# > 3# > 2#. It can be seen that sample 11# had the best effect on promoting blood flow of human-skin microcirculation, followed by 10#, and 2# was the weakest.

To more intuitively observe the blood-flow promoting effect of the 12 samples on the skin microcirculation of 10 healthy men and the individual differences, a broken line diagram was made according to the test results of the blood-flow promoting multiple of the 12 samples, as shown in Figure 5. It can be seen that due to the differences in body shape, physiological function, and other aspects of the 10 healthy adult male subjects, the data obtained from the blood-flow test on the human-surface microcirculation of 12 seamless knitted samples of each individual also had certain differences.

The two-factor variance analysis method in SPSS software (software version number: IBM SPSS Statistics 25.0) was used to analyze and study the blood-flow promotion multiple and arithmetic mean value of each sample on human-skin microcirculation in Table 4 and to explore the influence of the two factors of veil type and tissue structure on the blood-flow promotion multiple of each sample on human-skin microcirculation. As shown in Table 5, the inter-subject effect test results of the blood-flow promotion multiple of human-surface-skin microcirculation show that the adjoint probability P (0.000025) of FA was less than 0.05, while the adjoint probability P (0.949) of FB was more than 0.05, indicating that different levels of control variable A (veil type) had a significant influence on the observed variables. However, the control variable B (organizational structure) had no significant effect. It was concluded that different types of veils have significant differences in the blood-flow promotion multiple of human-skin microcirculation, but there was no significant difference in the tissue structure.

The S-N-K method was used to make a pairwise comparison of the differences between each level of the veil type, as well as between each level of the organizational structure, and the results are shown in Table 6 and Table 7. Based on the analysis of Table 6, conclusions can be obtained as follows: The higher the content of graphene in the veil yarn, the better the promoting effect of the fabric on the blood-flow rate of the human-surface-skin microcirculation. The four levels of the veil types were respectively in four different subsets. The first subset was the veil type GP-0.0%, in which the promotion multiple of the fabric woven by this veil type on human-surface-skin blood flow was small. The fourth subset was the veil type GP-0.8%, in which the fabric had a greater promotion multiple of the blood flow. There was only one level in each subset, and the four levels of the veil types had significant differences in the effect of promoting blood flow in human-surface-skin microcirculation.

Based on the analysis of Table 7, conclusions can be obtained as follows: When the tissue structure is 1+1 simulated ribbed, the fabric can promote the blood-flow rate of human-surface-skin microcirculation by a larger factor, followed by the weft plain stitch; while the 1+3 simulated ribbed fabric was poor. The organizational structures of the three levels were distributed in the same subset; there was no statistical difference.

The outline drawings of the estimated marginal mean value of the blood-flow promotion multiple of the skin microcirculation on the human surface are shown in Figure 6. Figure 6a shows that the promotion effect of fabrics corresponding to different veil types on the blood-flow velocity of the skin microcirculation on the human surface was significantly different. With the increase in graphene content, the promotion effect of fabrics on the blood-flow velocity of the skin microcirculation on the human surface was enhanced. As shown in Figure 6b, there was no significant difference in the tissue structure. The 1+1 simulated rib fabric had a better effect on promoting the blood-flow rate of skin microcirculation on the human surface, followed by plain weft knitted fabric, while 1+3 simulated rib fabric was weak.

## 4. Discussion

This paper mainly studied the effect of samples of different types of veils and different tissue structures on promoting the blood flow of human-skin microcirculation. Specifically, the promotion effect of 12 samples woven by different types of veils and different tissue structures according to the comprehensive experimental design method on the blood flow of human-skin microcirculation was tested. The test results of relevant indicators were studied and analyzed. The conclusions are as follows:Through testing the promotion times of 12 samples woven by different types of veils and different tissue structures according to the comprehensive experimental design method on the blood flow of human-skin microcirculation, the results show that the order of the promotion effect of each sample on the blood flow of human-skin microcirculation was: 11# > 10# > 12# > 8# > 9# > 7# > 5# > 6# > 4# > 1# > 3# > 2#. The effect of 11# sample on the blood flow of human-skin microcirculation and its promotion ability was the best;In the test of promoting multiple blood flow on human-skin microcirculation, the type of veil had a significant effect on the observed variables, while the tissue structure had no significant difference. The higher the content of graphene in the veil yarn, the better the promotion effect of the fabric on the blood flow of the human-skin microcirculation. The fabric made of the veil type GP-0.8% had the best promotion effect on the blood flow of the human-skin microcirculation. The order of the promotion effect of the veil type on the blood flow of the human-skin microcirculation was: GP-0.8% > GP-0.2% > GP-0.1% > GP-0.0%. When the tissue structure was 1+1 simulated rib, the promotion multiple of the fabric on the blood flow of the human-surface-skin microcirculation was large. The order of the promotion effect of the tissue structure on the blood flow of the skin microcirculation was 1+1 simulated rib > weft plain stitch > 1+3 simulated rib.

According to the material characteristics and manufacturing process of graphene fiber, this paper combined the functionality of graphene fiber, the maneuverability of weaving on the machine, and the wearability of the obtained fabric. It also measured the far-infrared performance of graphene fiber fabric used in the human body using the blood-flow velocity of the skin microcirculation on the human surface as an indicator [17,18]. It was concluded that the higher the content of graphene, the stronger its far-infrared functionality and the more obvious its role in promoting microcirculation on the human skin [19,20]. It was confirmed that graphene fiber has sufficient functionality and can be used as far-infrared medical care textiles in the medical auxiliary treatment of chronic pain diseases such as periarthritis of the shoulder [21]. However, because of the maneuverability of weaving of graphene fiber and the wearability of the fabric obtained, the graphene fiber selected in this experiment did not contain much graphene. We hope that based on this study, under allowable weaving conditions in the future, the graphene content in the fiber can be increased while ensuring the wearability of the fabric so that functional graphene fiber fabric can have a better application prospect in the field of far-infrared medical and health care textiles.

## Figures and Tables

**Figure 1 materials-15-06853-f001:**
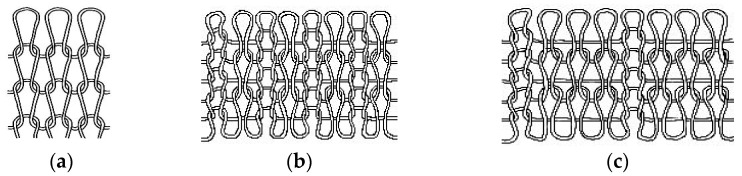
(**a**) The flat stitch; (**b**) the 1+1 simulated ribbed stitch; (**c**) the 1+3 simulated rib stitch.

**Figure 2 materials-15-06853-f002:**
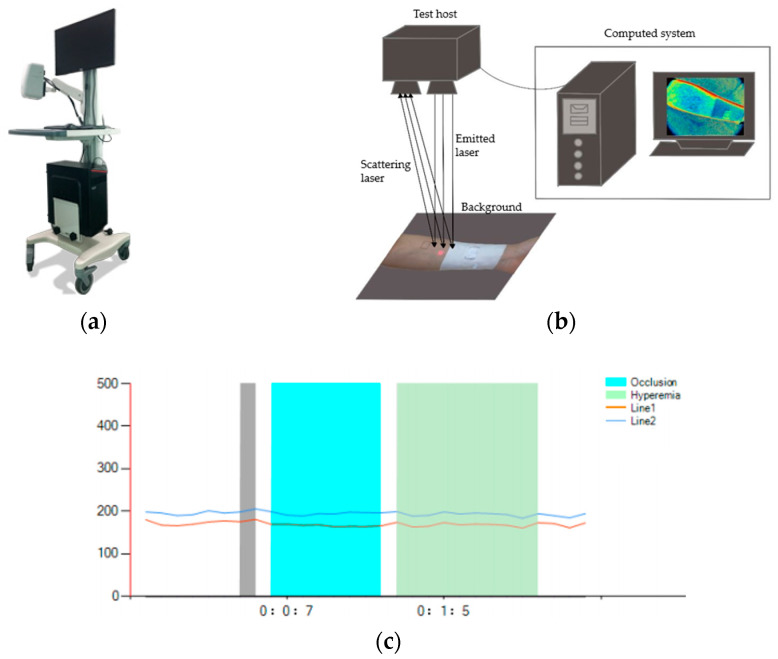
(**a**) The BVI microcirculation-blood-flow imager; (**b**) schematic diagram for measuring the blood-flow velocity of skin microcirculation on the surface of human body; (**c**) curves of blood-flow velocity of skin microcirculation on the surface of human body of a subject.

**Figure 3 materials-15-06853-f003:**
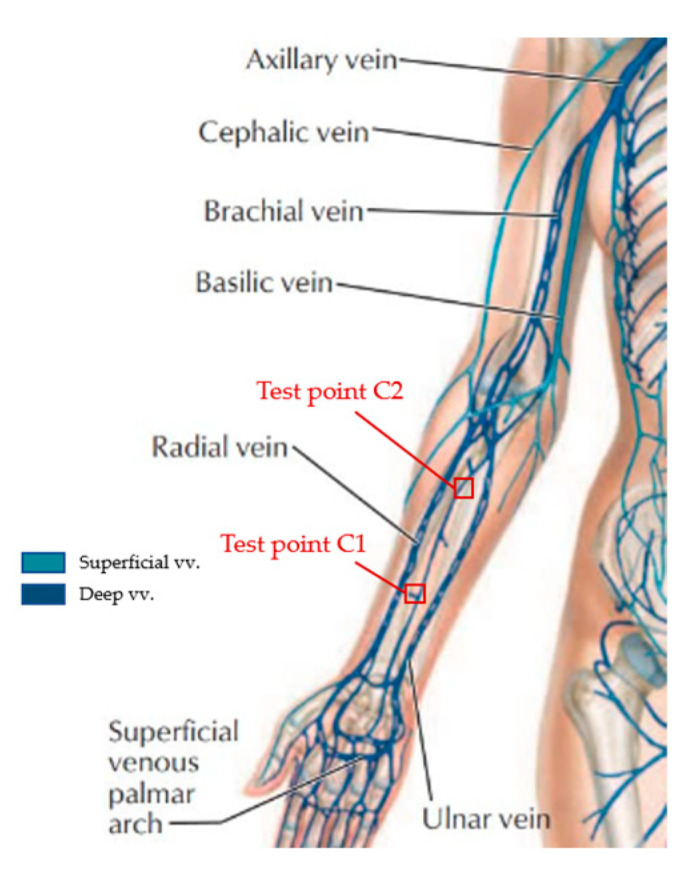
The diagram of major veins of the human arm and position marking of the arm test point.

**Figure 4 materials-15-06853-f004:**
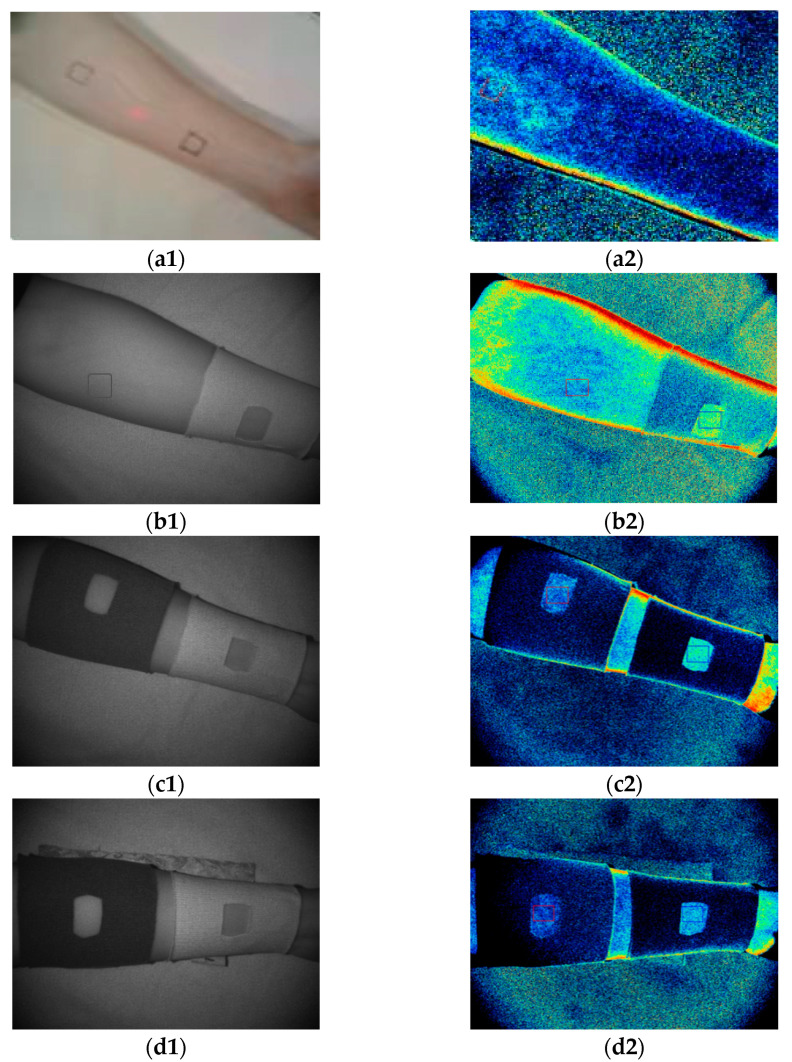
(**a1**) When the sample was not covered; (**a2**) blood-flow imaging when the sample was not covered; (**b1**) when GP-0.0% sample was covered; (**b2**) blood-flow imaging when covering GP-0.0% sample; (**c1**) when GP-0.1% sample was covered; (**c2**) blood-flow imaging when covering GP-0.1% sample; (**d1**) when GP-0.2% sample was covered; (**d2**) blood-flow imaging when covering GP-0.2% sample; (**e1**) when GP-0.8% sample was covered; (**e2**) blood-flow imaging when covering GP-0.8% sample.

**Figure 5 materials-15-06853-f005:**
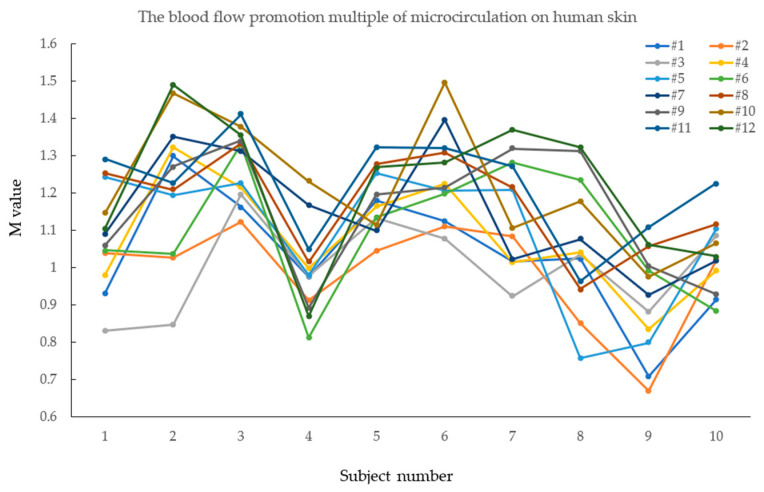
Fold-line diagram of promoting multiple blood flow on human-skin microcirculation of 12 seamless knitted fabrics.

**Figure 6 materials-15-06853-f006:**
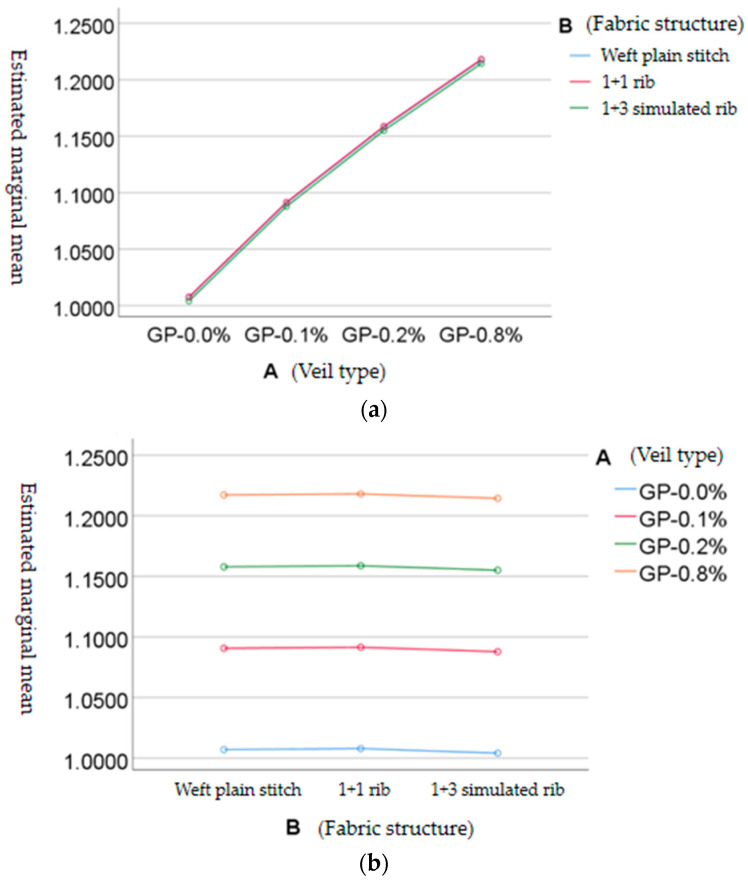
(**a**) Profile of estimated marginal mean value of blood-flow promotion multiple of human-skin microcirculation with independent variable A (veil type); (**b**) profile of estimated marginal mean value of blood-flow promotion multiple of human-skin microcirculation with independent variable B (fabric structure).

**Table 1 materials-15-06853-t001:** Veil materials and specifications.

Content	Yarn Type	Fineness	Supplier
GP-0.0%	Polyamide filament	77.8dtex(70D)	Yiwu Huading Polyamide Co., Ltd. (Yiwu, China)
GP-0.1%	Graphene polyamide filament	77.8dtex(70D)	Hangzhou Gaoxi Technology Co., Ltd. (Hangzhou, China)
GP-0.2%	Graphene polyamide filament	77.8dtex(70D)	Changzhou Henglibao Nano New Material Technology Co., Ltd. (Changzhou, China)
GP-0.8%	Graphene polyamide filament	77.8dtex(70D)	Shandong Shengquan New Material Co., Ltd. (Jinan, China)

**Table 2 materials-15-06853-t002:** Sample scheme sheet.

Fabric Number	A (Veil Type)	B (Fabric Weave)
1#	GP-0.0%	Weft plain stitch
2#	GP-0.0%	1+1 simulated rib
3#	GP-0.0%	1+3 simulated rib
4#	GP-0.1%	Weft plain stitch
5#	GP-0.1%	1+1 simulated rib
6#	GP-0.1%	1+3 simulated rib
7#	GP-0.2%	Weft plain stitch
8#	GP-0.2%	1+1 simulated rib
9#	GP-0.2%	1+3 simulated rib
10#	GP-0.8%	Weft plain stitch
11#	GP-0.8%	1+1 simulated rib
12#	GP-0.8%	1+3 simulated rib

**Table 3 materials-15-06853-t003:** The average values of dynamic blood-flow rate of skin microcirculation on the surface in three periods at two test points of a subject.

		Average	stdDev	Points	Min	Max	Median
**Roi1**							
	First Period	213.56	0.88	4256	212.86	214.80	213.01
	Second Period	214.78	2.25	4256	209.33	218.45	214.88
	Third Period	215.19	3.95	4256	207.98	221.03	214.06
**Roi2**							
	First Period	158.59	1.21	5504	156.92	159.77	159.77
	Second Period	160.48	3.05	5504	155.94	167.18	163.61
	Third Period	160.80	3.03	5504	156.16	166.85	161.67

**Table 4 materials-15-06853-t004:** Blood-flow promotion multiple and arithmetic mean value of each sample to skin microcirculation.

Sample Number	Tested Person										Average Value
	1	2	3	4	5	6	7	8	9	10	
1#	0.9297	1.2983	1.1624	0.9755	1.1792	1.1244	1.0155	1.0236	0.7079	0.9141	1.0331
2#	1.0377	1.0271	1.1224	0.9113	1.0446	1.1109	1.0841	0.8508	0.6684	1.0198	0.9877
3#	0.8305	0.8471	1.1951	0.9767	1.1317	1.0772	0.9232	1.0339	0.8810	1.0851	0.9981
4#	0.9789	1.3220	1.2150	0.9976	1.1631	1.2246	1.0141	1.0410	0.8345	0.9920	1.0783
5#	1.2433	1.1933	1.2274	0.9803	1.2526	1.2051	1.2081	0.7565	0.7988	1.1045	1.0970
6#	1.0459	1.0369	1.3295	0.8119	1.1351	1.1970	1.2809	1.2341	0.9908	0.8835	1.0946
7#	1.0903	1.3509	1.3130	1.1666	1.0981	1.3953	1.0215	1.0760	0.9267	1.0177	1.1456
8#	1.2529	1.2088	1.3328	1.0149	1.2768	1.3076	1.2149	0.9420	1.0573	1.1170	1.1725
9#	1.0594	1.2708	1.3407	0.8921	1.1949	1.2139	1.3186	1.3117	1.0042	0.9288	1.1535
10#	1.1457	1.4670	1.3779	1.2314	1.1140	1.4954	1.1059	1.1768	0.9760	1.0656	1.2156
11#	1.2908	1.2271	1.4108	1.0493	1.3214	1.3199	1.2716	0.9633	1.1087	1.2258	1.2189
12#	1.1042	1.4900	1.3553	0.8700	1.2687	1.2818	1.3691	1.3219	1.0602	1.0303	1.2151

**Table 5 materials-15-06853-t005:** Inter-subject effect test of blood-flow enhancement multiple of human-surface-skin microcirculation.

Dependent Variable: Promotion Multiple of Blood Flow on Human-Skin Microcirculation						
The Source	Type III Sum of Squares	Degrees of Freedom	The Mean Square	F	Significance	Partial Eta Square
Modified Model	0.074 ^a^	5	0.015	51.936	0.000	0.977
Intercept	14.986	1	14.986	52,908.098	0.000	1.000
A (Type of Veil)	0.074	3	0.025	86.525	0.000025	0.977
B (Histological Structure)	2.992 × 10^−5^	2	1.496 × 10^−5^	0.053	0.949	0.017
Error	0.002	6	0.000			
Total	15.061	12				
Total after Correction	0.075	11				
Modified Model	0.074 ^a^	5	0.015	51.936	0.000	0.977

^a^ Power of R = 0.977 (power of R after adjustment = 0.959).

**Table 6 materials-15-06853-t006:** Comparison of different types of veils on promoting multiple blood flow on skin microcirculation.

Blood-Flow Promotion Multiple Human-Surface-Skin Microcirculation					
S-N-K ^a,b^					
A (Veil Type)	Number of Cases	Subset			
		1	2	3	4
GP-0.0%	3	1.006300			
GP-0.1%	3		1.089967		
GP-0.2%	3			1.157200	
GP-0.8%	3				1.216533

**Table 7 materials-15-06853-t007:** Comparison of tissue structure in promoting multiple blood flow on skin microcirculation.

Blood-Flow Promotion Multiple Human-Surface-Skin Microcirculation		
S-N-K ^a,b^		
B (Histological Structure)	Number of Cases	Subset
		1
1+3 simulated rib	4	1.115325
Weft plain stitch	4	1.118150
1+1 simulated rib	4	1.119025
Significance		0.949

## Data Availability

All data can be found within the manuscript.

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
