# Peer review of "Study and Analysis of the Effect of Polyamide Seamless Knitted Fabric with Different Graphene Content on the Blood-Flow Velocity of Human-Skin Microcirculation"

_materials, 2022, doi:10.3390/ma15196853_

Round 1

Reviewer 1 Report

The scientific paper “Study and Analysis of the Effect of Polyamide Seamless Knitted Fabric with Different Graphene Content on the Blood Flow Velocity of Human Skin Microcirculation” aimed to explore the influence of seamless knitting samples prepared from polyamide with different graphene content and different tissue structures on the blood flow velocity of human surface skin microcirculation. It can be considered that:

1)      In Abstract: “In this paper, graphene polyamide yarns were combined with seamless knitting technology. To explore the influence of seamless knitting samples prepared from polyamide with different graphene content and different tissue structures on the blood flow velocity of human surface skin microcirculation”. Adjust this content into an introduction that provides greater contextualization of the problem and makes the objective of the study more clear.

2)      The caption of figure 2 is poor in content. It should provide more elements that avoid re-reading the manuscript text (methodology). Please complement.

3)      Figure 4 is difficult to visualize and understand. Please also adjust with a more complete caption.

4)      The way the authors present the manuscript is very confusing. I suggest the traditional formatting with introduction, materials and methods, results, discussion and conclusions.

5)      I do not consider 16 references a sufficient amount for the scientific basis of the research carried out. Very low number. Thus, it can be noted that it did not discuss its results with the literature.

Reviewer 2 Report

It is a nice experimental work on carbon reinforced fibers. I have only some minor comments that has to be modified before I can recommend it to be accepted.

1) page 1 line 24 - paper does not provide theoretical references - it is mainly experimental study - please correct

2) Introduction citing is not scientific way - first sentence citation should be behind entire sentence, that is, "shoulder joint [1]." Similarly, when starting with Wang then no need his first name just Wang et al. [3]... etc.

Proof reading by someone with advanced English proficiency would be also useful. Some sentences are hard to understand and I suggest simplification of them (e.g., p. 2 line 55 sentence is too long and hard to read; p. 2 line 59-60 sentence does not have active verb; p. 10 line 306-309 needs be simplified, etc.)

As to the material and used techniques, I only suggest to add or modify figure in such a way that reader`s can see where graphene within stitch is located. Figure 2, please also add some experimental measurements to this Figure not only as in original work as the separate Figure (moreover, presented schematic is too general and should be explained in more details (figure legends and descriptions should be improved). Figure 3 is not needed in manuscript can be just discuss in text, please remove.

Scientific soundness is acceptable and topic is of practical importance. Namely, work is of interest to community and after careful English editing and also once Figures are improved, then it would be worth being accepted. A key reason is that smart textile and/or novel materials for textile are now of great importance to community.

Reviewer 3 Report

Paper “Study and Analysis of the Effect of Polyamide Seamless Knitted Fabric with Different Graphene Content on the Blood Flow Velocity of Human Skin Microcirculation” by Zhang et al. explains the efficacy of graphene polyamide yarns combined with seamless knitting on the blood flow velocity of human surface skin microcirculation. The manuscript is well written and contains promising data. The authors achieved significant efforts in the manuscript, but it needs major revision before considering to be published in a materials journal.

1-     The abstract should be containing promising data to increase the significance of the manuscript.

2-     Please add a clear hypothesis at the end of the introduction.

3-     Please follow up the standard sequence according to journal style (Introduction, Material and Method, Results, Discussion, and Conclusion).

4-     The manuscript needs deep discussion and compares the obtained data with related studies.

5-     In my opinion, the number of samples is low to give accurate conclusions.

6-     The numbers in table 3 can be summarized into two decimal numbers.

7-     The number of citing references in the manuscript is very low, therefore the MS needs deep discussion with similar literature.

8-     What about the statistical analysis for the obtained data.

9-     The manuscript should be carefully grammatical and typo-error revision.

Round 2

Reviewer 1 Report

No comments

Reviewer 3 Report

Thank you for authors revision. The manuscript can be accepted for publication in the present form